# Glycine Alleviated Intestinal Injury by Inhibiting Ferroptosis in Piglets Challenged with Diquat

**DOI:** 10.3390/ani12223071

**Published:** 2022-11-08

**Authors:** Xiao Xu, Yu Wei, Hongwei Hua, Huiling Zhu, Kan Xiao, Jiangchao Zhao, Yulan Liu

**Affiliations:** 1Hubei Key Laboratory of Animal Nutrition and Feed Science, School of Animal Science and Nutritional Engineering, Wuhan Polytechnic University, Wuhan 430023, China; 2Department of Animal Science, Division of Agriculture, University of Arkansas, Fayetteville, AR 72701, USA

**Keywords:** ferroptosis, glycine, intestinal mucosa, oxidative stress, weanling piglets

## Abstract

**Simple Summary:**

Oxidative stress may reduce the growth performance and intestinal health status of weanling piglets. Due to the fact that the body can synthesize glycine, it is generally treated as an amino acid which is nonessential for nourishment. However, previous research has demonstrated that synthesized glycine was unable to support piglets’ newborn growth and development. Moreover, according to several findings, glycine is crucial for relieving oxidative stress and intestinal damage. The purpose of this investigation was to determine whether glycine could lessen the intestinal damage caused by diquat in weanling piglets and the relationship between ferroptosis and diquat-induced intestinal epithelial cell death. The results showed that dietary glycine reduced intestinal oxidative stress induced by diquat in weanling piglets. Furthermore, with increasing anti-oxidative capacity, dietary glycine was able to restrain intestinal epithelial cell ferroptosis triggered by diquat.

**Abstract:**

The purpose of this research was to examine the impact of glycine on intestinal injury caused by oxidative stress in piglets. A 2 × 2 factorial experiment with diets (basic diet vs. 1% glycine diet) and oxidative stress (saline vs. diquat) was conducted on 32 weanling piglets. On day 21, all piglets received an injection of either saline or diquat. After 7 days, all pigs were slaughtered and intestinal samples were collected. Dietary glycine supplementation improved intestinal mucosal morphology, increased the activities of disaccharidases and enhanced intestinal mucosal antioxidant capacity, while regulating the expression of ferroptosis mediators in the piglets under oxidative stress. These findings suggested that dietary glycine supplementation improved the morphology and function of the intestinal mucosa, which was involved in regulating antioxidant capacity and ferroptosis.

## 1. Introduction

Changes in dietary components, the use of medications and vaccines, and mycotoxin contamination of feeds, among other things, may result in the excessive formation of reactive oxide species (ROS), which causes oxidative stress in pigs in the current intensive swine production [1]. In particular, intestinal damage can result from severe oxidative stress [2]. The primary sources of ROS generation are found in abundance in intestinal epithelial cells’ mitochondria [3]. In addition to inducing apoptosis and preventing cell proliferation, ROS also interferes with intestinal function and retards intestinal development [4,5]. Therefore, dietary regulation is crucial to reducing intestinal damage brought on by oxidative stress.

The most recent type of cell death to be discovered is ferroptosis, which has recently been linked to oxidative stress [6,7]. The major features of ferroptosis are the buildup of iron ions in cells, the decreased ability of glutathione peroxidase 4 (GPX4) to repair lipid peroxidation injury, and the oxidation of polyunsaturated fatty acids including phospholipids [8]. Ferroptosis cells exhibit morphological traits such as ruptured mitochondrial outer membranes, reduced or absent mitochondrial cristae, and damaged cell membrane integrity [6]. In terms of biochemistry, ferroptosis may cause glutathione to be depleted and GPX4 activity to decline [4].

Glycine is typically viewed as a nutritionally unnecessary amino acid because the body can produce it [9]. However, a wealth of data has demonstrated that synthesized de novo glycine was unable to support piglets’ neonatal growth and development [10,11,12]. More and more studies have recently claimed that glycine is vital for relieving oxidative stress and liver damage [13,14,15]. Glycine is an important part of the antioxidant glutathione peroxidase (GSH-PX), which is a crucial regulator of ferroptosis [16]. Glycine may, therefore, have the ability to alleviate ferroptosis. 

In order to induce intestinal oxidative stress and damage in the weanling piglets, the pigs received an intraperitoneal injection of diquat. Intraperitoneal injection of diquat is a common and mature model to establish oxidative stress [17]. Some studies showed that an injection of diquat induced a reduction in productive performance, organ injury and increased production of ROS and so on [18,19]. The goal was to determine if glycine might enhance intestinal health by modulating the anti-oxidative capability and ferroptosis signaling pathway in piglets’ intestinal mucosa.

## 2. Materials and Methods

### 2.1. Experimental Animals and Design

An animal trial was conducted according to the Animal Scientific Procedures Act 1986 (Home Office Code of Practice. HMSO: London January 1997) and EU regulation (Directive 2010/63/EU). The whole procedure was approved by the Animal Care and Use Committee of Wuhan Polytechnic University (Wuhan, China). A total of 32 weanling piglets purchased from a commercial herd (Wuhan Charoen Pokphand Co., Ltd., Wuhan, China) with the following breeds-Duroc, Landrace, and Large White-with ages of 28 days and initial body weights (BW) of 7.18 ± 0.70 kg were employed in this study. Piglets were housed separately in 1.80 × 1.10 m^2^ stainless steel metabolic cages with unrestricted access to food and water in a climate-controlled environment. According to the specifications of the National Research Council, the experimental basal diet was designed [20].

A 2 × 2 factorial trial was used in the design of this investigation. Following a 21-day feeding of either a basal diet or one containing 1% glycine, all pigs received intraperitoneal injections of diquat (dibromide monohydrate, Chem Service, West Chester, PA, USA), either at a dosage of 10 mg/kg BW in saline or the same amount of saline. Diet type (basal or glycine diet) and oxidative stress were the determining variables for treatment (diquat or saline).

### 2.2. Sample Collection

All piglets were painlessly put to death by sodium pentobarbital injections at 80 mg/kg body weight a week after receiving diquat or saline injections. In accordance with our earlier work, the mid-jejunum and mid-ileum segments measuring 3 cm and 10 cm, respectively, were cut [21]. The fresh, 4% paraformaldehyde/PBS solution was used to gently flush the 3 cm intestine segments before storing them for histological observation [22]. The luminal chyme was gently washed out of the 10 cm intestinal samples after they were opened longitudinally. The mucosa samples were taken by scraping them off sterile glass slides, quickly freezing them in liquid nitrogen, and then storing them at −80 °C for analysis of the disaccharidases’ activities, the antioxidases’ activities, the protein, DNA, and RNA contents, and the levels of mRNA and protein expression.

### 2.3. Intestinal Morphology

The intestinal segments were fixed for 24 h, then dehydrated, embedded, and stained with hematoxylin and eosin. According to our prior research [23], a microscope (Olympus CX31, Japan) was used to measure the crypt depth and villus height at a magnification of 40×. At least 10 well-oriented and intact villi were chosen. Villus height was measured from the villus tip to the crypt mouth, and crypt depth was measured from the crypt mouth to the base.

### 2.4. Disaccharidase Activities of the Intestinal Mucosa

Using glucose kits, disaccharidase activities in the intestinal mucosa were measured with our prior investigation [22]. (No. A082-1 for lactase, No. A082-2 for sucrase, and No. A082-3 for maltase; Nanjing Jiancheng Bioengineering Institute, Nanjing, China).

### 2.5. Protein, DNA, and RNA Contents of the Intestinal mucosa

After homogenizing frozen mucosal samples, the supernatant was collected by centrifuging the mixture at 2500 rpm for 10 min. In accordance with earlier investigations, the supernatant’s protein, DNA, and RNA contents were examined [24].

### 2.6. Antioxidative Capacity of the Intestinal Mucosa

Total antioxidative capacity (T-AOC), activities of glutathione peroxidases (GSH-PX), contents of reductive glutathione (GSH), and malondialdehyde (MDA) of intestinal mucosa were determined using commercial kits from Nanjing Jiancheng Bioengineering Co. according to the previous study [1].

### 2.7. Gene Expression Analysis

The methods used in the previous study were followed for the isolation of total RNA, quantification, reverse transcription, and real-time PCR [23]. Table 1 displays the primer pairings for target gene amplification. Using the 2^−△△CT^ approach, the expression of the target genes in comparison to the housekeeping gene (glyceraldehyde-3-phosphate dehydrogenase; GAPDH) was examined. The piglets fed a basic diet and given saline injections were used to standardize the relative mRNA abundance of each target gene.

### 2.8. Protein Abundance Analysis

The methods for protein abundance analysis in intestinal mucosa were according to previous research [20]. Specific primary antibodies included rabbit anti-transferrin receptor protein 1 (*TFR1*, 1:1000; 86 kDa, #70R-50471; Fitzgerald, Rd. Sudbury, Acton, MA, USA), goat anti-solute carrier family 7 member 11 (*SLC7A11*, 1:1000; 55 kDa, #ab60171; Abcam, Cambridge, MA, USA), rabbit anti-glutathione peroxidase 4 (*GPX4*, 1:1000; 20 kDa, #10005258; Cayman Chemical Company, Rd. Ellsworth, Ann Arbor, MI, USA) and mouse anti-β-actin antibody (1:1000, 43 kDa, #A2228; Sigma-Aldrich, St. Louis, MO, USA). As a ratio of target protein/β-actin protein, the relative protein abundance of the target proteins (*TFR1*, *SLC7A11*, and *GPX4*) was expressed.

### 2.9. Statistical Analyses

Using the general linear model techniques (GLM) of SAS (SAS Inst. Inc., Cary, NC, USA), the data were analyzed as a 2 × 2 factorial experiment by ANOVA. The statistical model took into account the impacts of oxidative stress (saline or diquat), diet type (basal diet or glycine diet), and their interactions. Data were presented as means and SEMs. A post hoc analysis was carried out using Duncan’s multiple comparison tests when the interaction between diet and stress was significant. Differences were considered to be significant if *p* < 0.05. 

## 3. Results

### 3.1. Intestinal Morphology

According to Table 2, a significant interaction was observed, which was related to the interaction between diet and stress on the crypt depth of jejunum and villus height of ileum (*p* < 0.05). Dietary glycine significantly enhanced the crypt depth of jejunum in the piglets that suffered oxidative stress induced by diquat (*p* < 0.05). In addition, oxidative stress significantly decreased the ratio of the villus height to the crypt depth of ileum and jejunum, as well as the height of jejunal villus (*p* < 0.05). Compared with the piglets fed the control diet, the piglets fed dietary glycine had significantly enhanced villus height of ileum (*p* < 0.05).

### 3.2. Disaccharidases Activities of the Intestinal Mucosa

As shown in Table 3, a significant interaction existed between stress and diet for the activities of jejunal sucrase and maltase, and ileal maltase (*p* < 0.05). Compared with the piglets reared on the basal diet, the piglets reared with the glycine supplementation diet had increased the activities of jejunal mucosal disaccharidases of sucrase and maltase and enhanced the activity of ileal mucosal disaccharidases of maltase (*p* < 0.05).

### 3.3. Protein, DNA, and RNA Contents of the Intestinal Mucosa

As exhibited in Table 4, a significant interaction was presented between diet and stress on the contents of intestinal mucosal protein of the jejunum and ileum, ileal mucosal RNA/DNA, and protein/DNA (*p* < 0.05). Oxidative stress significantly decreased jejunal RNA/DNA and protein/DNA (*p* < 0.05). Overall, compared with the piglets treated with the control diet, the piglets treated with the glycine supplementation diet had increased the contents of protein of jejunum and ileum, and enhanced ileal RNA/DNA and protein/DNA (*p* < 0.05).

### 3.4. Antioxidative Capacity of the Intestinal Mucosa

Table 5 illustrated a significant interaction arisen in stress and diet for the activities of jejunal GSH-PX and GSH, MDA concentration of jejunum (*p* < 0.05), and the activity of ileal GSH (*p* < 0.05). Furthermore, oxidative stress inspired by injected diquat dramatically dropped the activity of T-AOC of jejunum, T-AOC, and GSH of ileum (*p* < 0.05). Compared with the piglets fed the basal diet, the piglets fed glycine supplementation significantly enhanced the activities of jejunal GSH-PX and GSH, increased the activity of GSH of ileum, and reduced the concentration of MDA of jejunum (*p* < 0.05).

### 3.5. Intestinal Mucosal Gene Expressions of the Key Genes Related to Ferroptosis

As displayed in Table 6, there was a significant interaction that was linked to stress and diet for the gene expressions of *TFR1*, *SLC7A11*, and *GPX4* of ferroptosis-related signals of jejunum in the piglets (*p* < 0.05). Compared with the piglets fed the control diet, the piglets fed glycine supplementation significantly improved the gene expression of jejunal *SLC7A11* and *GPX4* and reduced the gene expression of *TFR1* of jejunum (*p* < 0.05). Similarly, a significant interaction existed between stress and diet on the ileal gene expression of *TFR1*, *SLC7A11*, and *GPX4* of ferroptosis-related signals (*p* < 0.05). Moreover, oxidative stress triggered by injected diquat significantly increased the gene expression of *HSPB1* of ileum (*p* < 0.05). Interestingly, the glycine supplementation function in the ileum as in the jejunum about the gene expressions of ferroptosis-related signals in the piglets. Compared with the piglets reared on the basal diet, the piglets reared with glycine supplementation significantly prompted the gene expression of *SLC7A11* and *GPX4* of ileum and decreased ileal *TFR1* gene expression (*p* < 0.05).

### 3.6. Intestinal Mucosal Protein Abundance of the Key Proteins Related to Ferroptosis

A significant interaction existed between stress and diet on the protein abundance of TFR1 and GPX4 of jejunum of the key proteins related to ferroptosis in Figure 1. (*p* < 0.05). Compared with the piglets treated with the basal diet, the piglets treated with glycine supplementation significantly increased the protein abundance of jejunal GPX4 and decreased protein abundance of jejunal TFR1 (*p* < 0.05). Meanwhile, there was a significant interaction between stress and diet on protein abundance of ileal TFR1 (*p* < 0.05). Compared with the piglets fed the control diet, the piglets fed glycine supplementation significantly decreased the protein abundance of TFR1 of ileum (*p* < 0.05).

## 4. Discussion

This experiment aimed to investigate whether intestinal cells would undergo ferroptosis after the establishment of diquat-induced oxidative stress model in piglets, and to explore the protective effect and mechanism of glycine supplementation diet on intestinal injury in weaned piglets.

The morphological changes in intestinal tissue may be a direct reflection of how well the intestinal barrier is functioning. Intestinal VH, CD, and VH/CD are crucial indicators of intestinal mucosal morphology [25,26]. Disaccharides need to be further broken down into monosaccharides by intestinal mucosal cells (such as sucrase, lactase, and maltase) in order to be absorbed, because intestinal epithelial cells cannot do this directly. As a result, the activity of intestinal mucosal disaccharides is one of the key markers indicating intestinal digestive capacity [27,28]. Protein, RNA, and DNA are indicators of intestinal growth and development and damage repair. Protein/DNA and RNA/DNA can indicate the ability of protein synthesis [29,30]. In this study, oxidative stress induced by diquat stimulation significantly changed the morphological structure of intestinal villi, and decreased the activity of disaccharidase, protein content, RNA/DNA, and protein/DNA in piglets. Consistent with our research, Xiao et al. showed that oxidative stress can lead to impaired intestinal barrier function in piglets [31]. Xu et al. showed that oxidative stress can cause intestinal injury in piglets and significantly reduce the activity of intestinal digestive enzymes [32]. The above results were consistent with the results of this study. Interestingly, dietary glycine ameliorated intestinal morphological and structural damage, and increased disaccharidase activity, protein content, RNA/DNA, and protein/DNA, which demonstrated that glycine may be able to positively regulate intestinal structural and functional damage caused by oxidative stress in this study. Similar to our results, a variety of studies have shown that dietary glycine can improve the intestinal barrier function, and promote the villus growth of jejunum and ileum, which is beneficial to the development of small intestinal mucosa of piglets [30,33].

Intestinal antioxidant capacity is closely related to intestinal health; however, diquat can induce oxidative stress and decrease intestinal antioxidant capacity [34]. T-AOC displays the overall amount of total antioxidants in the body or organs, and MDA reflects the degree of peroxidation in the body, which is an essential marker of oxidative stress [35]. GSH is a cofactor of GSH-PX, which is a natural free radical scavenger in animals. GSH-PX plays a significant function in antioxidant damage and can accelerate the reduction of lipid peroxides [36]. In this study, it was found that under normal physiological conditions of piglets, dietary glycine significantly enhanced the activities of T-AOC in jejunum and GSH-PX in ileum. This may be because glycine is a vital precursor for the synthesis of GSH, and with the increase in dietary glycine, intestinal antioxidant capacity was enhanced. When piglets were exposed to oxidative stress induced by diquat, the activities of T-AOC and GSH-PX and GSH content in jejunum were significantly decreased, MDA content was significantly increased, and the activities of T-AOC and GSH-PX in ileum were significantly decreased. However, GSH-PX activity and GSH content in jejunum of piglets in the glycine supplementation group were significantly increased, MDA content was significantly decreased, and GSH content in ileum was significantly enhanced. These results suggested that glycine can enhance the antioxidant capacity and mitigate the damage of oxidative stress by increasing the activity of antioxidant enzymes in jejunum of piglets. Some studies have shown that supplementation of glycine in the diet can increase intestinal GSH content, enhance the intestinal antioxidant level and effectively relieve oxidative stress in mice [36,37]. Moreover, Hua et al. found that dietary glycine could increase the activities of T-AOC, GSH-PX, and GSH in the liver, reduce the content of MDA, and alleviate the oxidative stress induced by diquat injection in piglets [38]. This further illustrated that glycine may improve intestinal antioxidant capacity by increasing the GSH content [30,33].

Intestinal injury is inevitably accompanied by cell death; however, it is possible that intestinal injury brought on by oxidative stress has a distinct cell death mechanism rather than the usual classical cell death mode [39]. Ferroptosis is a non-apoptotic, iron-dependent cell death form that is intimately associated with the oxidative stress process. [6]. In this study, the mRNA and protein expression levels of intestinal ferroptosis-related signaling pathway were detected to explore its influencing mechanism. System Xc^−^/GPX4 signaling pathway is closely related to ferroptosis [40]. TFR1 acts as a carrier to transfer iron ions (Fe^3+^) into the cell’s inner membrane when cells undergo ferroptosis. HSPB1 is a molecular chaperone of several small heat shock proteins, which can reduce the concentration of Fe^3+^ by inhibiting the expression of TFR1, thereby alleviating the intensity of ferroptosis [41]. System Xc^−^ is composed of SLC7A11 and SLC3A2 subunits through disulfide bonds, which can transfer glutamate out of the cell through the cell membrane and cystine into the cell membrane at the same time. Finally, cystine is converted into cysteine and GSH is synthesized. GSH can reduce ROS under the action of the antioxidant enzyme GPX4 and alleviate oxidative damage to cells [42]. In this study, under diquat stimulation, the mRNA expressions of TFR1 and GPX4 in jejunum and TFR1 and HSPB1 in ileum were significantly increased, while the mRNA expression of SLC7A11 in ileum was significantly decreased, indicating that diquat stimulation caused a large amount of Fe^3+^ to enter cells, leading to oxidative stress in the intestine. At this time, the body alleviated oxidative stress injury by enhancing the activities of GPX4 and HSPB1. Meanwhile, dietary glycine significantly increased the mRNA expressions of SLC7A11 and GPX4 in jejunum and ileum, and significantly decreased the mRNA expression of TFR1 in jejunum and ileum after diquat stimulation. These results indicated that glycine can enhance the intestinal antioxidant system to inhibit the occurrence of ferroptosis, which is consistent with the results of intestinal antioxidant capacity. According to the results of mRNA expression and protein expression, glycine significantly reduced the expression of TFR1 protein in jejunum and ileum after diquat stimulation, and significantly increased the expression of GPX4 protein in jejunum. Consistent with the results of many studies, it was found that the expression of the GPX4 gene could inhibit iron death and alleviate cell damage [43,44,45]. Xu et al. have shown that dietary glycine can effectively inhibit the expression of TFR1, a key gene of ferroptosis, and promote the expression of GPX4, thereby alleviating liver injury induced by diquat [38]. These results suggested that dietary glycine can enhance the synthesis of GPX4, inhibit the occurrence of ferroptosis, and then protect the gut from the damage caused by ferroptosis.

## 5. Conclusions

In summary, the diet supplied with 1% glycine relieved intestinal damage by inhibiting the occurrence of ferroptosis and enhancing intestinal antioxidant capacity in the piglets under oxidative stress.

## Figures and Tables

**Figure 1 animals-12-03071-f001:**
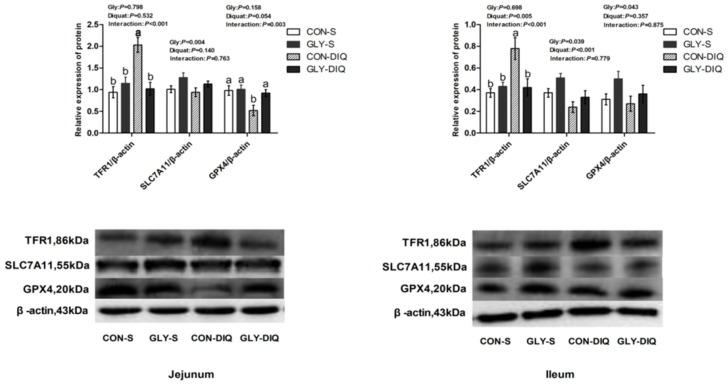
The abundance of ferroptosis-related proteins of jejunum and ileum in the piglets fed glycine supplementation diets under oxidative stress inspired by injected diquat. The stripes served as representational exemplars of Western blot images. Values are mean and pooled SEM, *n* = 8 (1 piglet per pen). CON-S, piglets fed the basal diet and injected with saline; GLY-S, piglets fed the glycine supplementation diet and injected with saline; CON-DIQ, piglets fed the basal diet and challenged with diquat; GLY-DIQ, piglets fed the glycine supplementation diet and challenged with diquat. a, b. No identical letter appears in the same row, which demonstrates that the difference is significant, *p* < 0.05.

**Table 1 animals-12-03071-t001:** Primer sequences used for real-time PCR.

Gene	Forward (5’-3’)	Reverse (5’-3’)	Annealing Temperature (℃)	Product Length (bp)	Accession Numbers
*TFR1*	CGAAGTGGCTGGTCATCT	TGTCTCTTGTCTCTACATTCCT	60	231	NM_214001.1
*HSPB1*	CTCGGAGATCCAGCAGACT	TCGTGCTTGCCCGTGAT	60	120	NM_001007518
*SLC7A11*	GCCTTGTCCTATGCTGAGTTG	GTTCCAGAATGTAGCGTCCAA	60	178	XM_021101587.1
*GPX4*	CTGTTCCGCCTGCTGAA	ACCTCCGTCTTGCCTCAT	60	218	NM_214407.1
*GAPDH*	CGTCCCTGAGACACGATGGT	GCCTTGACTGTGCCGTGGAAT	60	194	AF_017079.1

*TFR1*, transferrin receptor protein 1; *HSPB1*, heat shock protein beta 1; *SLC7A11*, solute carrier family 7 member 11; *GPX4*, glutathione peroxidase 4; *GAPDH*, glyceraldehyde-3-phosphate dehydrogenase.

**Table 2 animals-12-03071-t002:** The intestinal morphology of the piglets fed glycine diets under oxidative stress.

Item	Saline		Diquat		SEM	*p*-Value		
	Basal	Glycine	Basal	Glycine		Diet	Stress	Interaction
Jejunum								
VH (μm)	255	284	216	268	9	<0.001	<0.001	0.052
CD (μm)	162 ^b^	167 ^b^	145 ^c^	176 ^a^	4	<0.001	0.192	<0.001
VH/CD	1.57	1.71	1.49	1.53	0.05	0.006	<0.001	0.301
Ileum								
VH (μm)	272 ^a^	281 ^a^	242 ^b^	277 ^a^	7	<0.001	0.004	0.034
CD (μm)	171	167	164	169	4	0.978	0.216	0.069
VH/CD	1.59	1.69	1.48	1.65	0.04	<0.001	0.022	0.479

*n* = 8 (1 piglet per pen). ^a,b,c^ No identical letter appears on the shoulder of the average in the same row, which demonstrates that the difference is significant, *p* < 0.05, and on the contrary, the presence of the same letter means no significant difference. CD, crypt depth; SEM, standard error of mean; VH, villus height; VH/CD, the ratio of villus height to crypt depth.

**Table 3 animals-12-03071-t003:** The activities of intestinal mucosal disaccharidases of the piglets fed glycine diets under oxidative stress (U/mg protein).

Item	Saline		Diquat		SEM	*p*-Value		
	Basal	Glycine	Basal	Glycine		Diet	Stress	Interaction
Jejunum								
Lactase	25.8	32.5	27.9	31.6	2.4	0.003	0.854	0.680
Sucrase	36.4 ^a^	35.3 ^a^	26.6 ^b^	41.5 ^a^	4.0	0.312	0.412	<0.001
Maltase	173 ^ab^	194 ^a^	158 ^b^	184 ^a^	16	0.188	0.067	0.041
Ileum								
Lactase	6.02	10.06	6.23	9.87	0.68	<0.001	0.689	0.850
Sucrase	15.4	13.5	12.1	14.4	1.3	0.425	0.553	0.136
Maltase	107 ^a^	104 ^a^	74 ^b^	121 ^a^	12	0.215	0.430	<0.001

*n* = 8 (1 piglet per pen). ^a,b^ No identical letter appears on the shoulder of the average in the same row, which demonstrates that the difference is significant, *p* < 0.05, and on the contrary, the presence of the same letter means no significant difference. SEM, standard error of mean.

**Table 4 animals-12-03071-t004:** The contents of intestinal mucosal protein (mg/g tissue), RNA/DNA, and protein/NDA (mg/μg) of the piglets fed glycine diets under oxidative stress.

Item	Saline		Diquat		SEM	*p*-Value		
	Basal	Glycine	Basal	Glycine		Diet	Stress	Interaction
Jejunum								
Protein	5.48 ^a^	5.70 ^a^	4.65 ^b^	5.82 ^a^	0.26	0.006	0.135	<0.001
RNA/DNA	6.48	7.14	5.08	6.21	0.47	0.014	<0.001	0.215
Protein/DNA	0.12	0.15	0.10	0.12	0.01	0.034	0.038	0.674
Ileum								
Protein	5.69 ^a^	5.80 ^a^	4.43 ^c^	5.01 ^b^	0.24	0.245	<0.001	0.014
RNA/DNA	4.14 ^a^	3.68 ^ab^	2.21 ^c^	3.32 ^b^	0.23	0.004	<0.001	<0.001
Protein/DNA	0.08 ^a^	0.07 ^a^	0.05 ^b^	0.07 ^a^	0.01	0.528	0.031	0.003

*n* = 8 (1 piglet per pen). ^a,b,c^ No identical letter appears on the shoulder of the average in the same row, which demonstrates that the difference is significant, *p* < 0.05, and on the contrary, the presence of the same letter means no significant difference. RNA/DNA, the ratio of RNA to DNA; Protein/DNA, the ratio of Protein to DNA; SEM, standard error of mean.

**Table 5 animals-12-03071-t005:** The intestinal mucosal antioxidative capacity of the piglets fed glycine diets under oxidative stress.

Item	Saline		Diquat		SEM	*p*-Value		
	Basal	Glycine	Basal	Glycine		Diet	Stress	Interaction
Jejunum								
T-AOC, U/mg protein	0.510	0.554	0.404	0.463	0.042	0.046	0.011	0.753
GSH-PX, U/mg protein	16.3 ^b^	28.0 ^a^	12.3 ^c^	16.6 ^b^	1.9	<0.001	0.022	0.014
GSH, mg GSH/g protein	27.9 ^a^	27.8 ^a^	18.2 ^b^	24.3 ^a^	2.0	0.025	<0.001	0.032
MDA, nmol/mg protein	1.45 ^b^	1.32 ^b^	2.61 ^a^	1.68 ^b^	0.26	<0.001	<0.001	0.011
Ileum								
T-AOC, U/mg protein	0.304	0.298	0.225	0.285	0.035	0.216	0.031	0.386
GSH-PX, U/mg protein	24.5	53.6	20.4	42.5	3.1	<0.001	0.017	0.524
GSH, mg GSH/g protein	13.8 ^bc^	15.4 ^b^	8.8 ^c^	22.4 ^a^	3.0	<0.001	0.418	<0.001
MDA, nmol/mg protein	1.80	1.63	1.91	1.67	0.16	0.421	0.753	0.535

*n* = 8 (1 piglet per pen). ^a,b,c^ No identical letter appears on the shoulder of the average in the same row, which demonstrates that the difference is significant, *p* < 0.05, and on the contrary, the presence of the same letter means no significant difference. GSH, reduced glutathione; GSH-PX, glutathione peroxidases; MDA, malondialdehyde; SEM, standard error of mean; T-AOC, total antioxidative capacity.

**Table 6 animals-12-03071-t006:** The intestinal mucosal gene expressions of ferroptosis-related signals of the piglets fed glycine diets under oxidative stress.

Item	Saline		Diquat		SEM	*p*-Value		
	Basal	Glycine	Basal	Glycine		Diet	Stress	Interaction
Jejunum								
*TFR1*	1.00 ^b^	0.85 ^b^	1.63 ^a^	1.06 ^b^	0.14	0.010	<0.001	0.020
*HSPB1*	1.00	0.86	0.88	1.00	0.14	0.938	0.871	0.254
*SLC7A11*	1.00 ^b^	1.46 ^a^	0.64 ^c^	1.44 ^a^	0.18	0.003	0.274	0.002
*GPX4*	1.00 ^d^	4.69 ^c^	7.57 ^b^	16.02 ^a^	1.10	<0.001	<0.001	0.040
Ileum								
*TFR1*	1.00 ^b^	0.99 ^b^	1.66 ^a^	1.20 ^b^	0.16	0.243	0.005	0.042
*HSPB1*	1.00	0.89	1.37	1.04	0.13	0.025	0.003	0.354
*SLC7A11*	1.00 ^a^	1.28 ^a^	0.35 ^b^	1.03 ^a^	0.15	<0.001	<0.001	0.028
*GPX4*	1.00 ^b^	1.05 ^b^	0.80 ^b^	1.84 ^a^	0.13	0.024	0.421	<0.001

*n* = 8 (1 piglet per pen). ^a,b,c,d^ No identical letter appears on the shoulder of the average in the same row, which demonstrates that the difference is significant, *p* < 0.05, and on the contrary, the presence of the same letter means no significant difference. *TFR1*, transferrin receptor protein 1; *HSPB1*, heat shock protein beta 1; SEM, standard error of mean; *SLC7A11*, solute carrier family 7 member 11; *GPX4*, glutathione peroxidase 4.

## Data Availability

The data presented in this study are available in the manuscript.

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
