# Peer review of "Glycine Alleviated Intestinal Injury by Inhibiting Ferroptosis in Piglets Challenged with Diquat"

_animals, 2022, doi:10.3390/ani12223071_

Round 1
Reviewer 1 Report
The manuscript (Animals-1969000) entitled “Glycine alleviated intestinal injury by inhibiting ferroptosis in piglets challenged with diquat” was concluded that dietary glycine reduced intestinal oxidative stress induced by diquat in weanling piglets. Furthermore, with increasing anti-oxidative capacity, dietary glycine was able to restrain intestinal epithelial cell ferroptosis triggered by diquat. This study was well designed and the results were benefit for the use of additives in weanling piglets feeding. I suggest the manuscript should be revised following the points below before its acceptance for publication.
1. Setion 2.2 Sample collection: L82-83 Please state the dose of sodium pentobarbital injected.
2. Section 2.3 Intestinal morphology:Please describe the details of the measurement criteria for villus height and crypt height.
3. In the Conclusions section: Please add the supplementary dose of glycine level in the diet.
Author Response
The point to point response letter was attached as a file as below.

Reviewer 2 Report
The authors prepared a well organised and a well written manuscript. I read carefully and found no typographical errors, or mistakes. No major points have been raised. Authors may only improve Introduction part, add some recent and modern references to show that diquat poisoning can be deleterious in all animals. It can provoke severe stress, so some more references and further expalanation on diquat poisoning could improve the presentation of the work.
Author Response

(The authors gave the same response as above.)

Reviewer 3 Report
The manuscript named “Glycine alleviated intestinal injury by inhibiting ferroptosis in piglets challenged with diquat” was in well experimental design and correct data analysis. The results demonstrated that dietary glycine supplementation improved the morphology and function of the intestinal mucosa, which is involved in regulating antioxidant capacity and ferroptosis. These findings are meaningful to the weanling piglets feeding and amino acids nutrition concept. I recommend this manuscript should be accepted for publication after the bellowing minor revisions:
Line 30, …which was involved in regulating…
Line 40, …ROS also interfere with intestinal function and retard intestinal development…
Line 58, …and damage in the weanling piglets,…
Line 69, …the following breeds-Duroc, Landrace, and Large White-with ages…
Line 73, delete “(2012)”.
Line 76, …feeding of either a basal diet or…
Line 106, …Total anti-oxidative capacity (T-AOC), activities of glutathione…
Line 145, …significantly decreased the ratio of the villus height…
Line 146, Compared with the piglets fed control diet, …
Line 157, Line 170-171, Line 186, Line 201, 209, Line 222, 226, same as above.
Line 170, 172, 249, 252, “protein/DNA”.
Table 5, move the units of the parameters into the table.
Line 277, GSH-PX.
Conclusion section should be simplified to one or two sentences.
Author Response

(The authors gave the same response as above.)
